# OpenReview forum: "MEDIC: Comprehensive Evaluation of Leading Indicators for LLM Safety and Utility in Clinical Applications"
_TMLR — Accepted by TMLR_

### Review · Reviewer_aroK · 2026-02-03

**Summary Of Contributions:**

1. Proposes MEDIC, a “leading-indicators” evaluation framework for clinical LLMs, organized around five dimensions: medical reasoning, ethical & bias concerns, data/language understanding, in-context learning, and clinical safety.


2. Builds a heterogeneous task suite + matched protocols, explicitly mixing deterministic evaluation where possible (e.g., execution accuracy for SQL; exact/tolerance-based scoring for calculations) with structured proxy evaluation for open-ended generation.

3. Reports empirical “capability gaps” that static leaderboards miss, notably (i) a knowledge–execution gap between MCQ-style knowledge benchmarks vs operational tasks (MedCalc/EHRSQL), and (ii) a passive vs active safety gap (refusal vs error detection/correction)

**Audience:**

Yes

**Audience Explanation:**

TMLR readers who care about evaluation methodology, reliable LLM deployment, and domain-specific benchmarking would likely be interested.:

It makes a broadly relevant methodological point: static MCQ leaderboards can saturate and fail to predict functional performance, and “safety compliance” can differ from “error-detection ability.”

It contributes a concrete evaluation construct (CEF) that is reference-free and decomposes fidelity into interpretable components, which is of interest beyond healthcare (summarization, grounded generation, agent auditing).

**Broader Impact Concerns:**

No, the paper does not have to attract broader impact concerns.

**Claims And Evidence:**

Yes

**Claims Explanation:**

1. Knowledge–execution gap: The paper gives a clear operational definition (knowledge benchmarks vs operational tasks) and directly supports it with distributional evidence: knowledge tasks show “near-saturation” (median > 75%) while operational tasks have much lower medians (< 40%) under higher variance.

2. Passive vs active safety divergence: The argument is backed by explicit evaluation framing (Med-Safety refusal vs MEDEC error detection/localization) and a described sharp performance degradation from refusal saturation to near-zero error-detection performance for some models.

3. CEF provides a different signal than lexical metrics: The paper reports negligible correlations between CEF sub-scores and BLEU/ROUGE/BERTScore on a dataset with references (ACI-Bench), including the correlation table/values.

4. Open-ended ranking robustness: The authors mitigate position bias via bidirectional evaluation, use ~31k matches, and show very high agreement across three different judge models. This is a strong piece of evidence for ranking stability (even if not absolute correctness).

**Requested Changes:**

1. Provide sufficient details to replicate: full prompts, decoding settings, LLM-as-a-judge prompts, and post-processing; ideally an anonymized code release or at least an “evaluation card” appendix.

---

> ### Author Response · Authors · 2026-03-13
> **Methodological Transparency and Resource Accessibility**
>
> We thank the reviewer for the positive assessment of our work and the constructive feedback regarding reproducibility. We agree that providing comprehensive details is essential for the community to verify and build upon our findings. We have revised the manuscript and appendices to address the requested changes:
>
> - **Prompts:** We have included all relevant generation prompts and LLM-as-a-judge prompts in **Section A.6**.
>
> - **Decoding Settings:** The specific decoding parameters (temperature, top-p, top-k, and sampling strategies) for both the judge models and response generation are now detailed in **Table 4 (Appendix A.4.2)** and **Table 5 (Appendix A.4.3)**, respectively.
>
> - **Code, Resources, and Post-processing:** We added a **Resources** section (**Appendix A.5**) detailing model, code, and data availability, including links to the anonymized code repositories for our Cross-Examination Framework (CEF) and HealthBench. Regarding post-processing, we confirm that we do not apply manual corrections or external processing beyond the logic provided in these scripts. The only exception is a `max_tries` parameter used to retry generation in instances where the model output could not be parsed programmatically.
>
> - **Leaderboard:** To maintain double-blind anonymity, we cannot currently provide the URL to the live leaderboard. However, we have added **Appendix A.8**, which contains screenshots of the leaderboard interface. The full live version will be released upon publication.
>
> We believe these additions provide sufficient detail to reproduce our results. Please let us know if there is anything else we can do to improve the paper.

---

> ### Author Response · Authors · 2026-04-16
> **Additional Revisions and Validation Analyses**
>
> Following the receipt of all three reviews, we would like to inform the reviewer that we have made additional revisions to the manuscript for consistency. These include reinstating a clinician validation analysis and adding CEF human-validation results to the main text. We describe these additions in our responses to the other reviewers.

---

### Review · Reviewer_5Vrs · 2026-02-23

**Summary Of Contributions:**

This paper introduces MEDIC, a broad evaluation suite for clinical LLMs. It moves beyond USMLE-style QA and evaluates models across operational tasks (SQL, medical calculation), generative documentation, safety refusal, and error detection. The key empirical claims are:

1. Strong performance on static medical QA does not predict operational capability.

2. Passive safety (refusal) does not imply active safety (error detection).

3. Model performance is highly task-dependent; no single model dominates.

### Strengths

1. Important and timely problem.

2. Clear framing of knowledge–execution gap and passive vs. active safety.

3. Judge robustness analysis is thorough.

### Weaknesses

1. Heavy reliance on LLM-as-a-judge without validation against clinicians.

2. Judge agreement ≠ clinical validity.

3. CEF protocol may miss clinically critical error types.

4. Results section depends too much on the appendix.

**Audience:**

Yes

**Audience Explanation:**

The evaluation question is broadly relevant beyond healthcare. The knowledge–execution gap and safety findings are useful for the LLM evaluation community. However, the impact depends on addressing the validity concerns around automated judging and clinical grounding.

**Claims And Evidence:**

No

**Claims Explanation:**

Main concern: validity of LLM-as-a-judge The appendix explicitly describes using an LLM judge across multiple components, including Med-Safety scoring, open-ended QA evaluation, HealthBench rubric verification, and parts of summarization or note-generation evaluation, with Llama-3.1-70B-Instruct selected as the primary judge for consistency. The paper provides evidence that the open-ended QA Elo rankings are stable across different judge models, reporting near-perfect Spearman correlations. That is useful, but it only shows judge robustness, not judge validity. High agreement among judges can still reflect shared biases or shared failure modes, and does not demonstrate clinical accuracy. Since the work positions MEDIC as a set of leading indicators for clinical deployment, I need evidence that judge-based scores track human or clinician assessments of correctness and harm in medically meaningful ways. As it stands, I cannot tell whether MEDIC is measuring clinical quality or simply measuring what a particular judge model prefers.

Additional concerns

Results are hard to follow due to appendix dependence Even though the main paper includes helpful summary figures, I found it difficult to interpret model-level results and effect sizes while reading the Results section because key quantitative details are frequently deferred to the appendix.

Summarization faithfulness protocol may miss safety-critical errors The cross-examination setup includes a YES-only answers constraint for question generation with {YES/NO/IDK} verification. In clinical settings, many safety-critical mistakes are about negations, contraindications, dose and unit fidelity, and temporality. The paper should justify why this protocol captures those failure modes, or add targeted checks.

**Requested Changes:**

Questions for the authors

Do you have any human or clinician validation for the judge-based tasks? If not, why is it reasonable to treat judge scores as leading indicators for clinical safety?

You show judge stability for Elo rankings, but do you have evidence of correctness, not just consistency?

Why the YES-only constraint in cross-examination, and how does it handle negations and numeric or temporal errors?

---

> ### Author Response · Authors · 2026-03-13
> **Methodological Validation and Cross-Examination Design Decisions**
>
> We thank the reviewer for their assessment and for highlighting important areas for refinement within our framework. We agree that the validation of LLMs against clinicians is important. However, it is not the main objective of this paper. Our primary aim is to evaluate the performance of LLMs across diverse tasks under uniform settings to highlight their performance gaps, strengths, and weaknesses. While proper clinical validation studies for the specific judge tasks we utilize should be conducted, they fall outside the scope of this work. For evaluating models at scale, using an LLM as a judge is the most robust available alternative to surface-level lexical metrics like ROUGE, BLEU, or BERTScore. Existing literature supports this approach; for example, [Arora 2025 et al.] shows clinical alignment with GPT models for rubric assessment, and [Croxford 2025 et al.] demonstrates high inter-rater agreement between LLMs and clinicians in summarization tasks.
>
> For tasks involving open-ended QA pairwise evaluation to generate Elo ratings, we do not evaluate for clinical accuracy. Instead, we assess completeness, clarity, and safety. [Ramaswamy 2025 et al.] shows, during pairwise comparisons, the consistency with which it selects a model as the best in a matchup produces a metric that is 91% correlated with its own human-produced Elo score. While this study is not specific to the clinical domain, it supports our methodology since we do not measure clinical accuracy using open-ended QA.
>
> We placed individual task results in the Appendix to simplify the interpretation of the main findings, which are synthesized from different sets of results. If you could let us know which specific section is causing issues, we can move the relevant quantitative information to the main text to improve clarity.
>
> Regarding the YES-only constraint. We actually have another paper currently under review that investigates this choice through an extensive ablation study. The summary of our findings is as follows: our experiments demonstrate that mixed-answer prompts (YES/NO/IDK) lead to significant instability. We found that 22.3% of "IDK" questions and 5.6% of "NO" questions reverted to "YES" when re-evaluated against the source, whereas YES-only questions maintained 99.4% stability. Moreover LLMs struggle to generate verifiable negative claims without referencing unstated implications, making mixed-answer formats prone to noise. If the model misrepresents a number or omits a negation, it fails to support the "YES" answer, resulting in an "IDK" for omissions or a "NO" for explicit contradictions. Essentially, the YES-only constraint allows us to isolate the model's failure to preserve information without the confounding variable of unstable question generation
>
> Arora, R. K., Wei, J., Hicks, R. S., Chen, E., Parmar, N., & Kording, K. (2025). Evaluating large language models towards improved human health. _OpenAI Research_.
>
> Croxford, E., Gao, Y., First, E. _et al._ Evaluating clinical AI summaries with large language models as judges. _npj Digit. Med._ **8**, 640 (2025). https://doi.org/10.1038/s41746-025-02005-2
>
> Ramaswamy, A., Demeure, N., & Rrapaj, E. (2025, November). Model Consistency as a Cheap yet Predictive Proxy for LLM Elo Scores. In _Proceedings of the 2025 Conference on Empirical Methods in Natural Language Processing_ (pp. 30155-30163).

---

> ### Author Response · Authors · 2026-04-16
> **Revised Manuscript with New Clinician-Alignment Data**
>
> We thank the reviewer for the thorough assessment. Having received all reviews, we revisited our earlier response and provide a revised, point-wise response with new evidence and manuscript revisions.
>
> **1. LLM-as-a-judge validity: "I need evidence that judge-based scores track human or clinician assessments"**
>
> > "...I cannot tell whether MEDIC is measuring clinical quality or simply measuring what a particular judge model prefers."
>
> We address this with three complementary lines of evidence:
>
> (a) *Direct clinician comparison.* To validate the preferences of LLMs, we conducted a focused study in which 73 open-ended clinical questions were scored independently by both a practising clinician and multiple LLM judges on three clinically relevant axes. Mean error differences between a clinician and model-judge scores are shown in Table 4. On a 1-5 scale, these differences are relatively small and indicate strong alignment with clinical expert judgment. Across all models and evaluated axes, the mean scoring deviation remained tightly bounded between -1.14 and 0.45. This analysis is included in the revised manuscript (Appendix A.1 & Table 4).
>
> (b) *Supporting literature.* Arora et al. (2025) demonstrate clinical alignment with GPT models for rubric assessment, and Croxford et al. (2025) report high inter-rater agreement between LLMs and clinicians in summarization tasks.
>
> (c) *Multi-judge robustness.* Three independent judges show Spearman rho > 0.98 (Figure 5), confirming rankings are not artifacts of one judge's preferences.
>
> We believe that the convergence of clinician agreement (a), literature support (b), and judge invariance (c) collectively argues against the concern that MEDIC merely measures judge preference.
>
> **2. Correctness, not just consistency, of Elo rankings**
>
> > "...do you have evidence of correctness, not just consistency?"
>
> We address correctness and consistency separately:
>
> *Judge correctness:* The clinician comparison above directly assesses agreement between judge and clinician scores under an absolute scoring regime. The small mean differences across all three axes provide evidence that the judge is correct in an absolute sense, not merely self-consistent.
>
> *Pairwise comparison (Elo):* The pairwise judge evaluates clinical response quality along five defined criteria: relevance, completeness, safety, ethics, and clarity (Appendix A.7.10), aggregated across approximately 31,000 matches. Because the clinician comparison validates the judge's clinical assessment on related axes, Elo correctness follows from the demonstrated validity of the judge on the dimensions it evaluates. Additionally, Ramaswamy et al. (2025) show pairwise LLM-judge consistency produces Elo scores 91% correlated with human-produced Elo ratings. We have clarified this reasoning in Section 3.5.
>
> **3. YES-only constraint in cross-examination**
>
> > "...why the YES-only constraint, and how does it handle negations and numeric or temporal errors?"
>
> An independent validation of CEF for general-purpose evaluation (reproduced in Appendix [B] of the revised manuscript; per double-blind policy, the full citation is deferred to the camera-ready version) conducted an extensive ablation of this design choice. YES-only questions are explicitly grounded in the source: to generate one, the model must anchor it to a verifiable fact. NO questions allow introducing artifacts not in the source; IDK questions invite vague formulations. The ablation confirms this: 22.3% of IDK and 5.6% of NO questions reverted to YES when re-evaluated, whereas YES-only questions maintained 99.4% stability.
>
> Regarding negations and numeric errors: if the source states "administer 500mg" and the summary states "50mg," a YES question grounded in the source ("Is the dosage 500mg?") produces a NO or IDK answer when cross-examined against the summary, correctly flagging the error. The human validation (Appendix [B]) confirms CEF captures semantically meaningful errors at a 78.4% alignment rate with expert-annotated semantic errors. The ablation and validation results are summarized in the revised manuscript.
>
> **4. Appendix dependence**
>
> > "...I found it difficult to interpret model-level results..."
>
> We have moved key results tables and task-level breakdowns into Sections 3.3-3.4 to reduce appendix dependence. These changes are marked in the revision.
>
> Collectively, three converging lines of evidence (clinician agreement data, independent CEF validation, multi-judge invariance) establish that MEDIC's scores track expert clinical assessment and are not artifacts of model preference. The revised manuscript makes this evidence chain transparent, adds the clinician comparison table, and summarizes the CEF ablation. We believe these revisions address the core concern about judge validity and welcome further feedback.

---

### Review · Reviewer_7XNs · 2026-04-07

**Summary Of Contributions:**

MEDIC, presented in this paper, is an evaluation framework that leverages twenty existing benchmarks to assess five dimensions of clinical LLM competence and evaluates fifteen open-weight models across this suite.

Models are assessed on static knowledge retrieval, general control (though admittedly an odd nomenclature for this category given that it's math tasks), clinical text generation, functional execution, open-ended clinical QA, and safety/audit tasks. The paper also contributes the Cross-Examination Framework (CEF), a reference-free method for scoring generative clinical text by generating yes/no questions from both source and summary, then cross-checking answers to produce coverage, conformity, consistency, and conciseness scores.


Key strengths

+ Overall, we do need more diverse benchmarks in the medical space, so I'm inclined to accept this with a toning down of claims.
+ The knowledge-execution gap finding (Figure 4a) is interesting and important. Even if the specific benchmarks chosen are somewhat arbitrary, it convincingly demonstrates that some benchmarks in isolation tell you not nearly enough about operationalizability.
+ The passive/active safety divergence ( Figure 4b) is also an important finding, showing that model creators need to take a broader view of safety.
+ The rank correlation heatmap (Figure 7) is also very useful for understanding which benchmarks might be cross-predictive of other benchmark values.
+ Reproducibility seems to be above average with lots of information included!
+ The study covers a broad set of task types that others typically don't.
+ The paper looks at robustness to judge selection, which is great and not that common!

Key weaknesses

+ Though the paper positions MEDIC as a novel framework, the core contribution is assembling existing benchmarks in a rather arbitrary way. Compare MedHELM (Bedi et al., 2025), which validated its taxonomy with clinicians, or HealthBench (Arora et al., 2025), which involved physicians for its benchmark criteria. The paper does not report any clinician involvement in designing the taxonomy, and the dimension assignments in Table 1 feel arbitrary in places. Now, not all benchmarks require clinician involvement, but some justificaiton would be warranted.
    + As a nit, calling Math tasks General Control doesn't seem right.
+ The paper also calls out traditional benchmarks as lacking: " They tend to rely either on
static multiple-choice questions (MCQs), which fail to capture the multi-step reasoning inherent in complex
general-purpose and clinical tasks (Griot et al., 2025), or on subjective LLM-as-a-Judge assessments that
lack reproducibility." -> but many of the subtasks in this very benchmark do the same thing!
+ The Cross-Examination Framework is presented as a key methodological contribution but the paper could do more to vet that it is actually measuring something useful. Maybe with a small human validation? Right now, it says there's litte correlation with traditional metrics but then CEF is right and traiditonal metrics are wrong. But there's not evidence why CEF isn't wrong. This should be made more clear.
+ Also there is a claim that "cross examination framework helps detect hallucinations" but there is no verification of this claim. (Please correct me if I am wrong, though if I missed it, it would be great to make this a bit more clear in the text.)
+ The "leading indicator" language throughout the paper overstates what the evidence shows. A leading indicator, in its standard meaning, predicts a future outcome. The paper never tests whether MEDIC scores predict deployment success, clinical pilot outcomes, or any downstream measure of utility. The authors acknowledge this in the limitations, but the title, abstract, and introduction all frame the contribution in terms of "leading indicators" without this caveat. The leading indicator language becomes a bit more accurate as pertaining to cross-benchmark indicators of success (e.g., knowledge-execution gap). If that is the leading indicator claim being made, it should be more clear.
+ The paper cites Pimentel et al. (2024), who found evaluation harness choice alone can cause performance variation. It should be called out that this is a limitation or potentially should test two or three prompt variants per task would help establish whether the reported rankings are stable.

**Audience:**

Yes

**Audience Explanation:**

Overall, we do need more diverse benchmarks in the medical space, so I'm inclined to accept this with a toning down of claims. Also see above strengths:

+ Overall, we do need more diverse benchmarks in the medical space, so I'm inclined to accept this with a toning down of claims.
+ The knowledge-execution gap finding (Figure 4a) is interesting and important. Even if the specific benchmarks chosen are somewhat arbitrary, it convincingly demonstrates that some benchmarks in isolation tell you not nearly enough about operationalizability.
+ The passive/active safety divergence ( Figure 4b) is also an important finding, showing that model creators need to take a broader view of safety.
+ The rank correlation heatmap (Figure 7) is also very useful for understanding which benchmarks might be cross-predictive of other benchmark values.
+ Reproducibility seems to be above average with lots of information included!
+ The study covers a broad set of task types that others typically don't.
+ The paper looks at robustness to judge selection, which is great and not that common!


I particularly think Fig 7/4a/4b results are quite compelling and worth publishing!

**Broader Impact Concerns:**

I appreciated the warning about Goodhart's law in the impact statement. I think this is sufficient.

**Claims And Evidence:**

No

**Claims Explanation:**

From the above, there are a few spots where claims start to stretch what is backed. I'd be inclined to change this evaluation with one of: (1) some moderation of associated claims; (2) some more evidence supporting the claims.


+ Though the paper positions MEDIC as a novel framework, the core contribution is assembling existing benchmarks in a rather arbitrary way. Compare MedHELM (Bedi et al., 2025), which validated its taxonomy with clinicians, or HealthBench (Arora et al., 2025), which involved physicians for its benchmark criteria. The paper does not report any clinician involvement in designing the taxonomy, and the dimension assignments in Table 1 feel arbitrary in places. Now, not all benchmarks require clinician involvement, but some justificaiton would be warranted.
    + As a nit, calling Math tasks General Control doesn't seem right.
+ The paper also calls out traditional benchmarks as lacking: " They tend to rely either on
static multiple-choice questions (MCQs), which fail to capture the multi-step reasoning inherent in complex
general-purpose and clinical tasks (Griot et al., 2025), or on subjective LLM-as-a-Judge assessments that
lack reproducibility." -> but many of the subtasks in this very benchmark do the same thing!
+ The Cross-Examination Framework is presented as a key methodological contribution but the paper could do more to vet that it is actually measuring something useful. Maybe with a small human validation? Right now, it says there's litte correlation with traditional metrics but then CEF is right and traiditonal metrics are wrong. But there's not evidence why CEF isn't wrong. This should be made more clear.
+ Also there is a claim that "cross examination framework helps detect hallucinations" but there is no verification of this claim. (Please correct me if I am wrong, though if I missed it, it would be great to make this a bit more clear in the text.)
+ The "leading indicator" language throughout the paper overstates what the evidence shows. A leading indicator, in its standard meaning, predicts a future outcome. The paper never tests whether MEDIC scores predict deployment success, clinical pilot outcomes, or any downstream measure of utility. The authors acknowledge this in the limitations, but the title, abstract, and introduction all frame the contribution in terms of "leading indicators" without this caveat. The leading indicator language becomes a bit more accurate as pertaining to cross-benchmark indicators of success (e.g., knowledge-execution gap). If that is the leading indicator claim being made, it should be more clear.
+ The paper cites Pimentel et al. (2024), who found evaluation harness choice alone can cause performance variation. It should be called out that this is a limitation or potentially should test two or three prompt variants per task would help establish whether the reported rankings are stable.

**Requested Changes:**

See above:

Critical:

+ The Cross-Examination Framework needs some form of validation. The paper shows low correlation with ROUGE/BLEU/BERTScore and treats this as evidence that CEF is better, but as I noted above, there is no evidence that CEF is not also wrong. Even a small study would establish whether CEF scores track clinical quality judgments. Without this, the claim in Section 3.2 that the "cross examination framework helps detect hallucinations" is unverified. Related: the paper should test CEF sensitivity to the question-generation model, since all results currently depend on Llama-3.1-70B-Instruct as the evaluator (Appendix A.4.1). If I missed something, though, please do let me know. In that case, this should be more prominently featured in the main text.
+ The "leading indicator" framing needs to be either substantiated or walked back. A leading indicator predicts a future outcome. The paper never tests whether MEDIC scores predict deployment success, clinical pilot results, or any downstream measure. If the claim is narrower, about cross-benchmark predictiveness (e.g., the knowledge-execution gap), that should be stated explicitly. I could imagine a world where that's the leading indicator. Or at least some form of backing for this statement should be made.
+ The dimension assignments in Table 1 need some justification. Compare MedHELM, which validated its taxonomy with clinicians, or HealthBench, which involved physicians. Some assignments feel arbitrary. Additionally, calling math tasks "General Control" is confusing nomenclature that should be reconsidered.
+ The paper criticizes existing approaches for relying "either on static multiple-choice questions (MCQs), which fail to capture the multi-step reasoning inherent in complex general-purpose and clinical tasks, or on subjective LLM-as-a-Judge assessments that lack reproducibility." But MEDIC itself includes MCQ benchmarks (MedQA, MedMCQA, PubMedQA, MMLU-Pro) and uses LLM-as-a-Judge for Med-Safety, open-ended QA, HealthBench, and the CEF itself. Perhaps the sentence should be removed or reworded?

Recommended but not critical:

+ The paper cites Pimentel et al. (2024), who found evaluation harness choice alone can cause significant performance variation. Testing two or three prompt variants per task (at least for a subset of tasks) would help establish whether the reported rankings are stable or fragile to prompt wording. [This is nice to have, if not running the experiment, then acknowledging the limitation is important.]
+ Position more carefully against MedHELM and HealthBench in the main text (there's some discussion in the appendix but it's not sufficient).

---

> ### Author Response · Authors · 2026-04-16
> **CEF Validation Data, Methodological Revisions, and Benchmark Positioning**
>
> We thank the reviewer for the constructive feedback and for recognizing the value of the knowledge-execution gap, passive/active safety divergence, and rank correlation findings. We respond pointwise below.
>
> **1. CEF validation**
>
> > "...no evidence that CEF isn't wrong..."
>
> We address this with three complementary lines of evidence:
>
> (a) *Clinician agreement.* To directly validate the model-judge, we conducted a focused study in which 73 open-ended clinical questions were scored independently by both a practising clinician and LLM-judges across three clinically relevant axes (1-5 scale). Mean error differences between a clinician and model-judge scores are shown in Table 4. On a 1-5 scale, these relatively small differences confirm that LLM judges track clinical expert judgment rather than measuring their own preferences. Across all models and evaluated axes, the mean scoring deviation remained tightly bounded between -1.14 and 0.45. This analysis is included in the revised manuscript (Appendix A.1 & Table 4).
>
> (b) *Human validation of CEF.* An independent peer-reviewed validation of CEF for general-purpose evaluation (reproduced in Appendix [B] of the revised manuscript; per double-blind policy, the full citation is deferred to the camera-ready version) tested it against expert-annotated semantic errors, achieving 78.4% alignment, and against the FRANK hallucination benchmark (65.7% alignment). CEF sensitivity to the question-generation model was also tested across multiple judges using specialised metrics. These results are summarized in the main text and detailed in Appendix [B].
>
> (c) *Hallucination detection claim.* If CEF were merely measuring noise, we would expect poor alignment with expert annotations. The 78.4% rate confirms that CEF-flagged inconsistencies correspond to genuine semantic errors. The Consistency score detects factual misalignment through cross-examination failures (IDK answers), providing a mechanistic basis for the hallucination detection claim. We have revised Section 3.2 to make this evidence chain explicit.
>
> **2. "Leading indicator" framing**
>
> > "A leading indicator, in its standard meaning, predicts a future outcome."
>
> We appreciate this concern. We would note that our intended use of "leading indicator" is the simpler, literal one: an indicator of performance that can be measured upfront, before committing to costly deployment-based evaluation. In this sense it contrasts with lagging indicators (real-world clinical pilot outcomes, which only become available much later in the process). As we state in the limitations, MEDIC in no way substitutes for real-world testing; it provides an upfront proxy for model selection given that lagging indicators take months to materialize. The reviewer's observation that this framing "becomes a bit more accurate as pertaining to cross-benchmark indicators of success" is precisely our intent. We have revised the introduction and abstract to make this scope explicit. If the reviewer feels a different term would be clearer, we are open to adjusting the language further.
>
> **3. Table 1 assignments and "General Control"**
>
> We have added a rationale paragraph explaining the basis for each dimension assignment. Regarding "General Control": these benchmarks (AIME, GSM8K) monitor for catastrophic forgetting of general reasoning during clinical fine-tuning. We have renamed this category to "General Reasoning Controls" and clarified its purpose in the revised text.
>
> **4. MCQ/LLM-as-Judge self-contradiction**
>
> Our critique targets *exclusive reliance* on one evaluation modality, not the modalities themselves. MEDIC employs a hybrid strategy: deterministic metrics where tasks permit (EHRSQL, MedCalc, IFEval) and structured proxy evaluation (CEF, pairwise Elo) only for open-ended generation. We have revised the relevant sentence in the introduction to make this distinction clear.
>
> **5. Prompt sensitivity**
>
> The multi-judge robustness analysis (rho > 0.98, Figure 5) demonstrates ranking stability across adjudicators, and deterministic metrics for functional tasks are robust to prompt variation by design. That said, we agree this deserves explicit acknowledgment. We have added prompt sensitivity as a stated limitation per Pimentel et al. (2024).
>
> **6. MedHELM/HealthBench positioning**
>
> We have expanded the main-text discussion, incorporating relevant content from related work in Appendix A.2. The key differentiator is MEDIC's explicit decoupling of theoretical recall from operational agency, exposing the knowledge-execution gap that aggregated scores mask.
>
> These revisions strengthen the manuscript through converging evidence for CEF validity (clinician agreement, published human validation, judge invariance) and a more precise scope for MEDIC's contribution. The core findings remain robust under these refinements.

---

### Decision · Action_Editor_E12B · 2026-06-24

**Recommendation:** Accept as is

**Audience:**

Yes

**Audience Explanation:**

The paper addresses a timely and important problem in LLM evaluation, safety, and deployment. Its findings extend beyond healthcare and are relevant to researchers working on benchmarking, trustworthy AI, model evaluation, and LLM-based agents. The proposed evaluation framework and the reported capability gaps provide useful insights for both practitioners and researchers developing and assessing foundation models.

**Claims And Evidence:**

Yes

**Claims Explanation:**

The paper provides extensive empirical evidence across diverse benchmarks, tasks, and models to support its main conclusions. The findings on the knowledge-execution gap, the passive-versus-active safety gap, and task-dependent model performance are consistently demonstrated through quantitative analyses. While additional clinician validation of some judge-based evaluations would further strengthen the work, the evidence presented is sufficiently convincing for the paper’s stated claims and scope.